# Novel and Effective Parallel Mix-Generator Generative Adversarial Networks

## Abstract

In this paper, we propose a mix-generator generative adversarial networks (PGAN) model that works in parallel by mixing multiple disjoint generators to approximate a complex real distribution. In our model, we propose an adjustment component that collects all the generated data points from the generators, learns the boundary between each pair of generators, and provides error to separate the support of each of the generated distributions. To overcome the instability in a multiplayer game, a shrinkage adjustment component method is introduced to gradually reduce the boundary between generators during the training procedure. To address the linearly growing training time problem in a multiple generators model, we propose a method to train the generators in parallel. This means that our work can be scaled up to large parallel computation frameworks. We present an efficient loss function for the discriminator, an effective adjustment component, and a suitable generator. We also show how to introduce the decay factor to stabilize the training procedure. We have performed extensive experiments on synthetic datasets, MNIST, and CIFAR-10. These experiments reveal that the error provided by the adjustment component could successfully separate the generated distributions and each of the generators can stably learn a part of the real distribution even if only a few modes are contained in the real distribution.

## 1 Introduction

Generative Adversarial Networks were proposed by Goodfellow et al. (2014), where two neural networks, generator and discriminator, are trained to play a minimax game. The generator is trained to fool the discriminator while the discriminator is trained to distinguish fake data (generated data) from real data. When Nash Equilibrium is reached, generated distribution $P_G$ will be equal to the real distribution $P_{real}$. Unlike Restricted Boltzmann Machine (RBM, Salakhutdinov et al. (2007)) or Variational Auto-encoder (VAE, Kingma & Welling (2013)), that explicitly approximate data distribution, the approximation of GAN is implicit. Due to this property, training GAN is challenging. It has been reported that GAN suffers from the mode collapse problem (Goodfellow (2016),Metz et al. (2016)). Many methods have been proposed to solve this problem (Salimans et al. (2016), Srivastava et al. (2017), Grover & Ermon (2017),Arjovsky et al. (2017)). In this paper, we propose a new model to solve this problem.

Similar to the work of (Rasmussen (2000)), we use a set of generators to replace the single, complex generator. Each generator only captures a part of the real distribution, while the distance between the mix-generated distribution and the real distribution should be minimized. An adjustment component is added to achieve separation between each pair of generators, and a penalty will be passed to the generator if an overlap is detected. Moreover, we propose a shrinkage adjustment component method to gradually reduce the effect of the penalty, since the strict boundary will lead to a non-convergence problem. Practically, forcing each generated distribution to be totally disjoint will cause potential problems. More specifically, we observe two problems in practice: (1) competition: multiple generators try to capture one mode, but are hampered by a strict boundary. This happens when the total number of generators $K$ is greater than the actual number of modes of $P_{real}$. (2) One beats all: One or a few of the generators are strong enough to capture all the modes, while the other generators are blocked outside and capture nothing. To solve these problems, we propose the following approach: (1) use reverse KL divergence instead of JS Divergence as the generator loss, to reduce the generator's ability to capture all the modes, and (2) introduce a shrinkage adjustment

method to gradually reduce the weight of the adjustment component $C$ based on the training time and the difference between each generator loss. We will discuss the details in part 3. Benefiting from such design, there is no need to pre-define the number of generators, and stable convergence can be obtained when the new component shrinks to zero. Finally, our model can allow parallelized training among generators, with synchronized or asynchronized updated for the discriminator, which reduces the training time.

To highlight, our main contributions are:

1. In Sections 3.1 and 2, we propose a multi-generator model where each generator captures different parts of the real data distribution while the mixing distribution captures all the data.

2. We introduce an adjustment component to separate between generated distributions. The adjustment can work with any discriminator.

3. In Section 3.3, we propose a shrinkage component method which reduces the penalty to guarantee convergence. If the penalty shrinks to zero, we will minimize $D_{KL}(\sum_k^K \alpha_k G_k || P_{real})$.

4. We organize the shared memory to allow for parallel training to reduce the training time. Our algorithm scales well even on large parallel platforms.

5. In Section 4, we use synthetic and real data to illustrate the effectiveness of our design.

## 2 RELATED WORK

Recently, many researchers have started focusing on improving GAN. Arjovsky & Bottou (2017) show that the zero sum loss function will lead to a gradient vanishing problem when the discriminator is trained to be optimal. The heuristic loss function contains reverse Kullback Leibler divergence(KL divergence)(Nowozin et al. (2016)). Note that the reverse KL divergence ($KL(P_{model} || P_{data})$) has the property that $P_{model}$ will tend to capture a single mode of $P_{data}$, while ignoring the other modes.(Theis et al. (2015)). As a consequence, the reverse KL divergence term contained in the heuristic generator loss function will cause the mode collapse problem of GAN.

To solve the mode collapse problem, Metz et al. (2016) proposed unrolled GAN, where copies of the discriminator are made, and back-propagation is done through all of them, while the generator is updated based on the gradient update of discriminator. Srivastava et al. (2017) use another reconstructor network to learn the reverse mapping from generated distribution to prior noise. If the support of the mapped distribution is aggregated to a small portion, then the mode collapse is detected.

Nowozin et al. (2016) show that the discriminator loss of GAN is a variational lower bound of f-divergence. The maximization in minimax game is to approximate a tighter lower bound of f-divergence, while the minimization is to minimize the lower bound. Zhao et al. (2016) understand GAN as an energy model, where an autoencoder is used as the energy function, or, the discriminator learns the manifold of real distribution, and a hinge loss is adopted to illustrate the model. Further, Berthelot et al. (2017) extend the autoencoder discriminator to measure the Wasserstein distance. Note that in an energy based GAN, mode collapse doesn't occur if a proper energy function is shaped and the manifold of the data is well learned.

Arjovsky et al. (2017) use Wasserstein-1 distance or Earth Mover distance instead of f-divergence (Jensen Shannon Divergence or KullbackLeibler divergence). Wasserstein metric can always provide a non-zero gradient, and is stable during training. However, the k−Lipschitz condition has to be ensured. However, the truncated parameter doesn't work well and the training speed is relatively slow comparing to f-GAN. Arora et al. (2017) extend the wasserstein GAN to a multi generator scheme. They prove the equilibrium of mix vs one game and the condition needed to win the game.

Tolstikhin et al. (2017) use the idea of Adaboost (Freund et al. (1996)), where the weight of the misclassified data is increased, and the final model is a mixture of all the weak learners trained in previous steps. Mode collapse can also be resolved since the weight of non-captured data points will be increased. Nguyen et al. (2017) proposed a dual discriminator model where KL and reverse KL di-

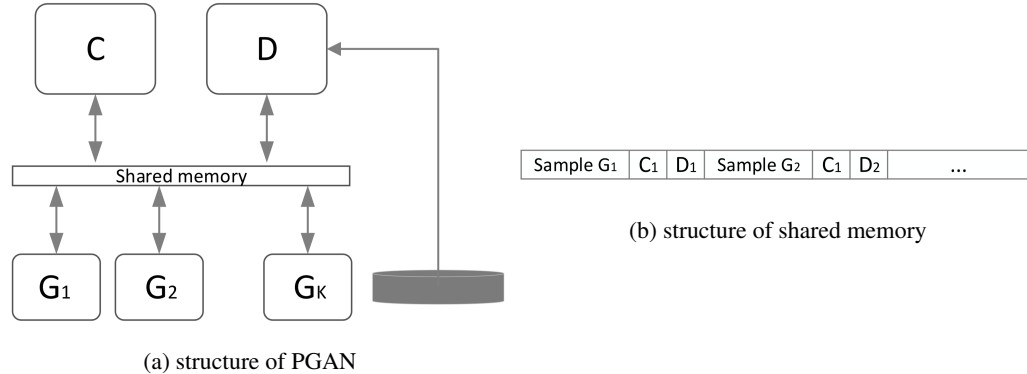

(a) structure of PGAN

(b) structure of shared memory

Figure 1: Illustration of proposed PGAN

vergence are controlled by two discriminators. Durugkar et al. (2016) propose a multi-discriminator model, where weak discriminators are trained using parts of the data, and the gradients from all the discriminators are passed to the generator.

## 3 OUR METHOD

In this section, we will show how our model is designed. We train $K$ generators, where the mix of the generators $G = \sum_k^K \alpha_k G_k$ approaches the real data distribution $P_{real}$. The support of each generated distribution $G_k$ is trained to be disjoint. To ensure that $G_k$ is disjoint (for each $1 \leq k \leq K$), we propose an adjustment component $C$ to dynamically classify $G_k$ and $G_{-k}$. A partition error is provided to each of the generators as an overlapping indicator.

### 3.1 STRUCTURE OF PGAN

We refer to our parallel model as PGAN. The structure of PGAN is shown in Figure 1. We use a set of simple generators to replace the original single complex generator. All the generators, the discriminator and the coprocessor are connected by shared memory. The communication between them only happens through the shared memory. The shared memory has $K$ slots, where $K$ is the number of generators. Each slot contains two parts: a sample part where the samples generated by the generator $k$ are stored, and a validation part where the value of the discriminator and the adjustment component are stored. Thus the total size of the shared memory is $k(batchsize + 2)$.

During training, generator $k$ will store its generated sample in the $kth$ slot, and wait for the response from $D$ and $C$. If the value in the validation part is updated, generator $k$ will update its parameter based on the value obtained from $D$ and $C$.

The discriminator and coprocessor, have two phases in one cycle: training phase and validation phase. During the validation phase, $D$ or $C$ will get the data points from the sample part of each slot, and execute the feed forward step. Note that the $kth$ batch error of $C$ and $D$ in the validation phase is exactly the validation $C_k$ and $D_k$ for generator $k$. During the training phase, $D$ or $C$ will get the corresponding data, and execute the backward step. The training and validation phases can be swapped in one batch, or in one epoch.

Note that the response time of the discriminator cannot be guaranteed when the number of generators is large. Assume that the forward step takes time $t_F$. In this case the back propagation time will be $a \times t_F$ practically (e.g., for NVIDIA Tesla $a > 3$ depends on the network size.). If $K > a$, there will be waiting time for the generator. To further reduce or eliminate the waiting time, duplicated discriminators are needed, specifically, one copy of the discriminator for every $a$ generators. The parameter for the discriminator can be either synchronized (periodically copy) or asynchronized (Hogwild training Recht et al. (2011)). Note that Hogwild training requires a sparse model and input. However, in our experiment, Hogwild training performs well even when no sparse regularization is applied.

### 3.2 LOSS FUNCTIONS

The loss function proposed by Goodfellow et al. (2014) with a heuristic generator loss is:

$$\mathcal{J}^D = \mathbb{E}_{x \sim P_{data}}[\log D(x)] + \mathbb{E}_{z \sim p_z(z)}[\log(1 - D(G(z)))]$$
$$\mathcal{J}^G = \mathbb{E}_{z \sim p_z(z)}[\log(D(G(z)))] \tag{1}$$

For each generator, the optimal discriminator is $D^* = \frac{P_{data}}{P_{data} + P_g}$. When convergence is reached, we obtain $P_g = P_{data}$.

In PGAN, we design a multi-player game by dividing one generator into K generators, and adding an adjustment component to the original loss function. The loss function for each generator is:

$$\mathcal{J}^D(G, D) = \mathbb{E}_{x \sim P_{data}}[\log D(x)] + \mathbb{E}_{x \sim P_G}[\log(1 - D(x))]$$
$$\mathcal{J}^C(G, C) = \mathbb{E}_{x \sim P_{g_{-k}}}[\log C(x)] + \mathbb{E}_{x \sim P_{g_k}}[\log(1 - C(x))] \tag{2}$$
$$\mathcal{J}^{G_k}(G_k, C, D) = \mathbb{E}_{x \sim P_{g_k}}[1 - \log(\frac{1}{D} - 1)] - \beta \, \mathbb{E}_{x \sim P_{g_k}}[\log(1 - C(x))]$$

Note that the last loss function $\mathcal{J}^{G_k}(G_k, C, D)$ is not bounded below. We have to truncate $\mathcal{J}^{G_k}$ if $D > t$ to avoid the gradient explosion problem, where $t$ is a threshold value.

The goal is to solve the minimax game in a multiple-player game:

$$\max_D \mathcal{J}^D, \max_C \mathcal{J}^C, \min_{G_k} \mathcal{J}^{G_k} \tag{3}$$

If we take a closer look at the loss functions, we notice that: (1) The discriminator loss $\mathcal{J}^D$ is nothing but the loss from the original GAN paper, which minimizes the Jensen-Shannon Divergence (JSD) between the mixture of generators and $P_{real}$. (2) the adjustment component loss $\mathcal{J}^C$ is actually another discriminator that treats $G_{-k}$ as real, $G_k$ as fake, and separates each generator $G_k$ from all the other generators $G_{-k}$ maximizing $JSD(G_k||G_{-k})$. (3) each generator is trained according to the gradient provided by both the discriminator $D$ and a weighted adjustment component $C$.

### 3.3 SHRINKAGE ADJUSTMENT COMPONENT

For a three players game, if we think of all the generators as one player, the result may not converge unless one player is out, since the convergence of two players is proved. In PGAN, we employ a weight factor $\beta$ in the adjustment component, and gradually decrease the weight $\beta$. When $\beta \to 0$, the three players game can be reduced back to a two players game, as in the original GAN, where the Nash Equilibrium could be reached.

At the beginning of the training procedure, $C$ is high enough to dominate the game, which means that no overlapping is allowed for the support of each generated distribution. With the training process going, the distance $D(\sum_k \alpha_k G_k || P_{real})$ will get saturated and will not decrease since the generators are strictly non-overlapping. We then gradually reduce the adjustment component to allow overlaps. As a consequence, the generated distributions that are close to each other will gradually merge, while those generators that are not close to each other will keep the distance due to the property of the reverse KL divergence.

There are several ways to choose $\beta$:

1. Set a constant value during training.
2. Decrease $\beta$ based on the number of iterations, i.e., $\beta = \exp^{-\lambda t}$, where $\lambda$ is a hyper parameter related to the total number of iterations.
3. Decrease $\beta$ based on the difference of the generator loss $\mathcal{J}^G$, i.e., $\beta = \sigma(\Delta \mathcal{J}^G)$
4. Combine 2 and 3.

The first method is a 'diversity factor' similar to Hoang et al. (2017), where $\beta$ is predefined to control the diversity of the generated distribution. However, constant $\beta$ may not converge when the the modes of the real distribution highly overlap, or the number of modes is less than the number of

generators. The second method is relatively stable but will cause a slow converge situation when the learning is faster than expected. The third method uses the strategy that when $\mathcal{J}^G$ becomes unstable, $\beta$ is decreased. The instability of the generator in this case is due to the one-beat-all problem. If the loss of the one generator is much higher than those of the others, this specific generator is forced out, leading to an oscillation of the generated distribution. So we prefer the last method, where both the number of iterations and the difference of the generator loss are considered.

## 3.4 DESCRIPTION OF PGAN

Algorithm 1 provides details on PGAN. The generator stores the generated sample in the shared memory. The discriminator and the adjustment component fetch the sample and return the corresponding error back to the shared memory. The generator fetches the error and does the back-propagation. As is discussed in 3.1, more discriminator and adjustment components are required to reduce the waiting time. In the worst case, if we assign each generator a copy of the discriminator, and a copy of the adjustment component, the total resources used will be $O(K)$, while the expected running time when using Hogwild training is $O(1/K)$. For a synchronize update, the running time will be $O(\frac{Nbatch}{K})$ if updated in each iteration. The worst case time can be reduced if a powerful discriminator is trained since before the synchronized update, the discriminator is strong enough to guide the generator and the generator can still learn something even if the parameters of the discriminator are not updated. Note that the running time for the adjustment component is negligible since the model complexity and data size of $C$ are small compared to the discriminator and the dataset.

---

**Algorithm 1** Algorithm for Training PGAN

---

1: K: number of processors
2: Niter: number of iterations
3:
4: Generator:
5: **for** $k = 0$ to $K$ **do**
6:     **for** $n = 0$ to $Niter$ **do**
7:         Sample $m$ samples from prior $P_g(z)$;
8:         Generate $m$ samples;
9:         Save sample in shared memory slot $k$;
10:        Wait for validation;
11:        **if** $D_k$ and $C_k$ are updated **then**
12:            Backprop according to $\mathcal{J}^G$;
13:            update $\beta$ based on $\mathcal{J}^D$ and $R_\beta$;
14:        **end if**
15:    **end for**
16: **end for**
17:
18: Discriminator:
19: **for** $n = 0$ to $Niter$ **do**
20:     **for** $k = 0$ to $K$ **do**

21:         Sample $m$ samples from $P_d$;
22:         Feedforward, obtain $real\ error$;
23:         Get $m$ samples from slot $k$;
24:         Feedforward, obtain $fake\ error$;
25:         Save $fake\ error$ to shared memory;
26:         Backprop according to $\mathcal{J}^D$
27:     **end for**
28: **end for**
29:
30: Adjustment Component:
31: **for** $n = 0$ to $Niter$ **do**
32:     **for** $k = 0$ to $K$ **do**
33:         Get $(k-1)m$ samples from slot $-k$
34:         Backprop according to $\mathcal{J}^C$
35:         Get $m$ samples from slot $k$;
36:         Feedforward, obtain error $C_k$;
37:         Save $C_k$ to shared memory;
38:     **end for**
39: **end for**

---

## 3.5 THEORETICAL ANALYSIS

In this section, we will show that the distance we are minimizing is $D_{KL}(P_{g_k}||P_{data})$ and $-D_{JSD}(P_{g_k}||P_{g_{-k}})$. From Goodfellow et al. (2014), the optimal discriminator given current generator $G$ has a close form $D_G^* = \frac{P_{data}(x)}{P_{data}(x)+P_g(x)}$. Since the loss function of $C$ is fairly close to $D$, we can obtain the optimal $C$ given that the current $G$ is $C_G^* = \frac{P_{G_{-k}}(x)}{P_{G_{-k}}+P_g(x)}$. Next, we will analyze the loss of the generator when we fix $D = D^*$ and $C = C^*$.

**Proposition 1** *Given optimal $D^*$ and $C^*$, minimizing the loss for generator in equation 2 is equivalent to minimizing:*

$$D(P_{g_k}, P_{data}, P_{g_{-k}}) = D_{KL}(P_{g_k}||P_{data}) - \beta D_{JSD}(P_{g_k}||P_{g_{-k}})$$

**Proof 1** *We first show that minimizing the first term is equivalent to minimizing $D_{KL}(P_{g_k}||P_{data})$. If we take the partial derivative of the reverse KL divergence:*

$$\frac{\partial}{\partial \theta} D_{KL}(P_{g_k}(\theta)||P_{data}) = \frac{\partial}{\partial \theta} \int P_{g_k}(\theta) \log \frac{P_{g_k}(\theta)}{P_{data}} \, dx.$$

*We can use Leibniz integral rule to switch integral and derivative, if assume that the function inside the integral satisfies: 1. continuity, 2. continuous derivative, and 3. $\lim_{x \sim \infty} f(x) = 0$. We obtain:*

$$\frac{\partial}{\partial \theta} D_{KL}(P_{g_k}(\theta)||P_{data}) = \int \frac{\partial P_{g_k}(\theta)}{\partial \theta} \log \frac{P_{g_k}}{P_{data}} + P_{g_k} \frac{\partial P_{g_k}(\theta)}{\partial \theta} \, dx.$$

*Substitute $D$ with optimal $D^*$, $\mathcal{J}^{G_k}(G_k, C, D)$ can also be rewritten as:*

$$\mathcal{J}^{G_k}(G_k, C, D^*) = \mathbb{E}_{x \sim P_G}[1 + \log(\frac{1 - D^*}{D^*})] = \mathbb{E}_{x \sim P_G}[1 + \log \frac{P_{g_k}(\theta)}{P_{data}}]$$

$$= \frac{\partial}{\partial \theta} \int \log \frac{P_{g_k}}{P_{data}} P_{g_k}(\theta) + P_{g_k}(\theta) \, dx = \int \log \frac{P_{g_k}}{P_{data}} \frac{\partial P_{g_k}(\theta)}{\partial \theta} + P_{g_k} \frac{\partial \log P_{g_k}(\theta)}{\partial \theta} \, dx,$$

*which is equivalent to the gradient of the reverse KL divergence. Note that we assume that $\frac{P_{g_k}}{P_{data}}$ is a constant when optimal $D^*$ is obtained.*

*The second term in the generator loss is the same as the zero-sum loss in Goodfellow et al. (2014), which is equivalent to minimizing the Jensen Shannon Divergence $D_{JSD}(P_{g_k}||P_{g_{-k}})$.* □

We can also show that by reducing $\beta$, the generator will only capture the mode that has been captured.

**Proposition 2** *When the adjustment component shrinks to zero, i.e. $\beta \to 0$, the gradient of the generator will vanish and the algorithm will converge.*

**Proof 2** *According to Proposition1, if $\beta \to 0$, $D_{JSD}(P_{g_k}||P_{g_{-k}}) \to 0$ and no gradient is provided. For the reverse KL divergence, if $P_{g_k} \to 0$, $D_{KL}(P_{g_k}||P_{data}) \to 0$, thus the gradient vanishes independent of if $P_{data}$ is nonzero or not. If $P_{g_k}$ is non-zero while $P_{data}$ is zero, the large gradient will push $P_{g_k}$ to zero.* □

From proposition2, we understand that the shrinkage adjustment component method is important to guarantee the convergence of the algorithm, even if not all the modes of $P_{data}$ were captured. The property of reverse KL divergence is also important to stabilize the model. By contrast, if we use JSD or KL Divergence, the non-zero $P_{data}$ will push up $P_{g_k}$ from zero, which breaks up the boundary between generators, and the goal of separating $P_{g_k}$ will fail.

## 4 EXPERIMENTS

In this section, we demonstrate the practical effectiveness of our algorithm through experiments on three datasets: synthetic datasets, MNIST, and CIFAR-10. In the case of synthetic datasets, we have employed two different datasets. In the first synthetic dataset, a standard 8 mixture of Gaussians is used as the target distribution to test the ability of our model to capture the separate modes. In the second datatset, we increase the number of generators to force a competition situation. Under competition, some generators will be forced out if the boundary is strict. We have noted that by introducing a shrinkage component, several generators can merge to one and achieve final convergence. The set up for all the experiments is: (1) Learning rate 0.0002, (2) Minibatch size 128 for generator, discriminator and adjustment components, (3) Adam optimizer (Kingma & Ba (2014)) with first-order momentum 0.5, (4) SGD optimizer for Hogwild trained discriminator, (4) $\beta$ is set to 1 at the begininng, with decay $\beta = \exp^{-\lambda t}$, and (5) Layers as LeakyReLU, weight initialization are from DCGAN (Radford et al. (2015)). All the codes are implemented by Pytorch.

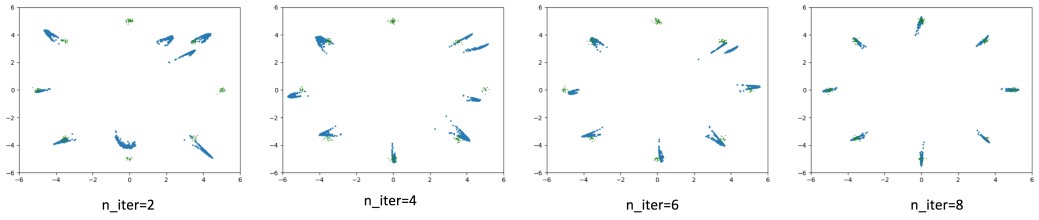

Figure 2: The case of 8 generators: green points refer to the real data distribution and the blue points form the generated distribution

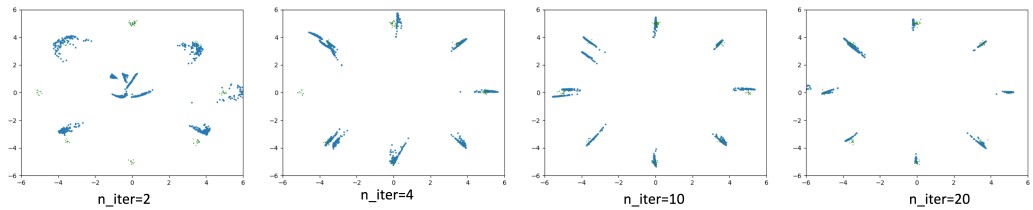

Figure 3: The case of 10 generators: green points form the real data distribution and the blue points form the generated distribution

## 4.1 SYNTHETIC DATASETS

We have generated two synthetic datasets to study the function of the adjustment component and the shrinkage method. We have constructed the generator, the discriminator and the adjustment component as three layer fully connected neural networks. The capacity is large enough for the demonstration.

The first dataset is a mixture of 8 Gaussians without any overlaps, as is shown in Nguyen et al. (2017) and Metz et al. (2016). First, we train exactly 8 generators with random initialization. In Figure 2, we show the results for every 5k steps (discriminator steps).

From these results we see that all the generators are spread out at the beginning. The penalty from the adjustment component is high and the generators proceeding in the same direction will be divided after certain number of steps. Since the number of modes is exactly the same as the number of generators, the property of the reverse KL divergence will keep each generator stay stationary even when $\beta$ becomes small. Finally all the 8 modes are captured by different generators.

We have then increased the number of generators to 10. This experiment is relevant since the number of modes may not be known in advance. The result is shown in Figure 3. At the beginning the situation is the same as in the previous setting, but the strong penalty will hamper the mode captured by two generators. The two generators are competing for the same mode. This illustrates that the function of the shrinkage component method is to mediate the competition between generators. However, $\beta$ cannot be small at the beginning, since it will hamper the separation function of the adjustment component.

## 4.2 REAL WORLD DATA

In this section, we use two popular datasets, MNIST(LeCun & Cortes (2010)) with 60,000 28 by 28 pixels hand written digits, and CIFAR-10(Krizhevsky et al.) with 50,000 32 by 32 pixels real images, to test the effectiveness of our model. Note that we scale up the MNIST dataset to 32. The network structure is similar to DCGAN. To evaluate the quality of generated samples, we use the Inception Score proposed in Salimans et al. (2016), where the score is calculated by the expectation of KL divergence $\mathbb{E}[D_{KL}p(y|x)||p(y)]$, where we calculate the distance between conditional label and real label. The score is a good indicator on the quality of generated images. More importantly, we the

use inception score to check the diversity of the image generated by single, and mixed generators. For inception score, we use the library from Tensorflow.

When training MNIST and CIFAR-10 datasets, we designed a relatively strong discriminator with a high learning rate, since the gradient vanish problem is not observed in reverse KL GAN. The update of the discriminator is synchronized, and Hogwild training is also tested, but the score is a little bit lower than for the synchronized case.

### 4.2.1 MNIST DATASET

The MNIST dataset contains 10 classes. We ran our model with different number of generators ranging from 5 to 15. The result is shown in fig 4. Note that by increasing the number of generators, the diversity score for each generator decreases, however, the diversity score for mixed generators is high. This dataset is simple since there are only 10 classes and hence we cannot actually observe an increasing diversity using the mix generator. The losses for all the generators are low enough and no generator is forced out. The inception score of the mixed generator is pretty high since the dataset is simple to capture. However, by increasing the number of generators, the score for each generator decreased, since the boundary limits the search space for a single generator. The decrease also gets saturated since we shrink the weight of the adjustment component, and overlaps are accepted.

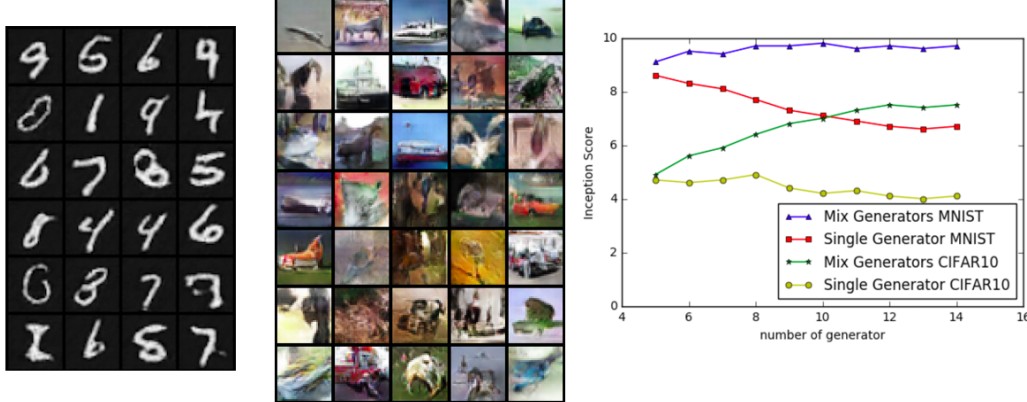

Figure 4: Left: Random pick from the mix generator for MNIST dataset. Mid: Random pick from the mix generator for CIFAR-10 dataset. Right:Inception score for mix generators and single generator for both datasets.

### 4.2.2 CIFAR-10 DATASET

CIFAR-10 dataset contains 10 labels, but the number of modes is much more than 10. We trained 10 to 20 generators for this dataset. From the results we can conclude that the diversity score increases with the number of generators, while it gradually gets saturated. From our observation, the threshold depends on the complexity of the dataset, model capacity, and the adjustment component. The inception score increases by increasing the number of generators, while it got saturated eventually. The highest score we get is 7.15, with more than 12 generators. For a single generator in the mixture, the score is relatively low due to the limitation of diversity. Note that the decrease of a single generator is smaller than what is in the MNIST dataset, since the search space for CIFAR-10 is much larger and the mix will not get saturated with a small number of generators.

### 4.3 TRAINING TIME

The training time for the sequential mix generator model for CIFAR-10 dataset is 115.4 min in our setting. To obtain the same score, the PGAN with 10 generators and Hogwild updated discriminators takes 51.6 mins, which takes only 44.7 percent of the sequential running time. And for synchronized updated discriminator, the running time is 61.7 min, which takes 53.5 percentage of the regular time.

The running time is still far from optimal (10 percent). For Hogwild training, the convergence rate is not gauranteed if the sparsity condition is not satisfied. For synchronized updating, the condition of optimal discriminator cannot be gauranteed, even though a more complex (both in capacity and learning rate) discriminator is adopted.

## 5 CONCLUSIONS

In this paper, we propose a mixed generator method to solve the mode collapse problem of GAN, and our algorithm is parallelizable, and can be scaled to large platforms. To conquer the competition and one-beat all problems in the mix generator model, we have designed the reverse KL divergence loss function, and an adjustment component decay to produce a stable, converging, and fast training method. The results show we can handle the situation when the generators compete for the same mode even when the number of generators is greater than the number of modes. The shrinkage method which gradually reduced extra component to zero will eliminate the adjustment player and reduce to multi-generator vs discriminator game.

More works need to be done in this multi-player game. First, the shrinkage method can also be improved if we can have a better heuristic for $\beta$. Or we can train to learn $\beta$, to achieve balance between competition and convergence. Second, the weight for each generator can also be dynamic. The generator learns more should have higher weight. Finally, new parallelization algorithm with less communication cost could be investigate to accelerate the multi-generator model since currently the run time is far from optimal.

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
