# OpenReview forum: "NOVEL AND EFFECTIVE PARALLEL MIX-GENERATOR GENERATIVE ADVERSARIAL NETWORKS"
_ICLR.cc/2018/Conference — Reject_

### Official Review · AnonReviewer2 · 2017-11-25
**Interesting idea to train parallel generators, but not ready for publication**

**Rating:** 3
**Confidence:** 5

**Review:**

Overall, the writing is very confusing at points and needs some attention to make the paper clearer. I’m not entirely sure the authors understand the material particularly well, as I found some of the arguments and narrative confusing or just incorrect. I don’t really see any significant contribution here except “we had this idea for this model, and it works”. There’s no interesting questions being asked about missing modes (and no answers through good experimentation), no insight that might contribute to our understanding of the problem, and no comparison to other models. My guess is this submission was rushed (and perhaps they were just looking for feedback). I like the idea, don’t get me wrong: a model that is trainable across multiple GPUs and that distributes generative work is pretty cool, and I want to see this work succeed (after a *lot* more work). But the paper really lacks what I’d consider good science, and I don’t see it publishable without significant improvement.

Personally I think you should change the angle from missing modes to parallel training. I don’t see any strong guarantees that the model will do what you say it will, especially as beta goes to zero.

Detailed comments

P1
“, that explicitly approximate data distribution, the approximation of GAN is implicit”
The wording of this is pretty strange: by “implicit”, we mean that we only have *samples* from the distribution(s) of interest, but what does it mean for an approximation to be “implicit”?

From the intro, it doesn’t sound like the approach is meant for the “mode collapse” problem, but for dealing with missing modes. These are different types of failures for GANs, and while there are many theories for why these happen, to my knowledge there’s no such consensus that these issues are the same.
For instance, what is keeping each of the generators from collapsing onto a single value? We often see the model collapse on several different values: why couldn’t each of your generators do this?

P2: No, it is incorrect that the KL is what is causing mode collapse, and I think actually you mean “missing modes”. Arjovsky et al addresses the mode collapse problem, which is just another word for a type of instability in GANs. But this isn’t because of “vanishing gradients”, as the “proxy loss” (which you call “heuristic loss”, this isn’t a common term, fyi), which is what GANs are trained on in practice don’t vanish, but show some other sorts of instabilities (Arjovsky 2016). That said, other GAN variants without regularization also show collapse *and* missing modes, such as LSGAN and all the f-GAN variants (even the auto encoder variants).

You should also probably cite Che et al 2016 as another model that addressed missing modes. Also, what about ALI, BiGAN, and ALiCE? These also address missing modes (at least they claim to).

I don’t understand why you’re comparing f-GAN and WGAN convergences: they are addressing different things with GANs: one shows insight into what exactly traditional GANs are doing (solving a dual problem of minimizing an f-divergence) versus addressing stability through using an IPM (though also a dual formulation of the wasserstein). f-GANs ensure neither stability nor non-vanishing gradients.

P3: I like the breakdown of how the memory is organized.
This is for multi-GPU, correct? This needs to be explicitly stated.

P6:
There’s a sign error in proof 1 (both in the definition of the reverse KL and when the loss is written out).
Also, the gradient w.r.t. theta magically appears in the second half.
This is a pretty round-about way to arrive at that you’re minimizing the reverse KL: I’m pretty sure this can be shown by formulating the second term in f-gan (the one where you sample from the generator), that is f*(T), where f* is the convex conjugate of f = -log

Mixture of Gaussians: common *missing modes* experiment.

So my general comments about the experiments
You need to compare to other models that address missing modes. Overall, many people have shown success with experiments similar to your simple mixture of Gaussians experiments, so in order to show something significant here, you will need to have a more challenging experiments and show a comparison to other models.
The real-world experiments are fairly unconvincing, as you only show MNIST and CIFAR-10 (and MNIST doesn’t look very good). Overall, the good inception scores aren’t too surprising given the model has several generators for each mode, but I think we need to see a demonstration on better datasets.

---

### Official Review · AnonReviewer3 · 2017-11-27
**Avoiding mode collapse in GANs through combination of multiple weak generators.**

**Rating:** 6
**Confidence:** 4

**Review:**

Summary:
This paper proposes parallel GANs (PGANs). This is a new architecture which composes the generator based on a mixture of weak generators with the main intended purpose that each unique generator may suffer mode collapse, but as long as each generator collapses to a distinct mode, the combination of generators will cover the whole image distribution. The paper proposes a number of technical details to 1) ensure that each sub generator offers distinct information (adjustment component, C) and 2) to efficiently train the generators in parallel while accumulating information to update both the discriminator and the adjustment component.
Results are shown on a synthetic dataset of gaussian mixtures, demonstrating that the model does indeed find all modes within the data, and on two small real image datasets: MNIST and CIFAR-10. Overall the parallel generator model results in ~x2 speedup in training time compared with a single complex generator model.

Strengths:
Mode collapse in GANs is a timely and unsolved problem. While most work aims to construct auxiliary loss function to prevent this collapse, this paper instead chooses to accept the collapse and instead encourage multiple models which collapse to unique modes. Though this does present a new problem in chooses the number of modes to estimate within a data source, the paper also presents a solution to systematically combine redundant modes over time, making the model more robust to the choice of number of generators overall.

Weaknesses:
Organization - The paper is quite difficult to read. Some concepts are presented out of order. For example, the notion of an adjustment component is very natural but not introduced until after it is mentioned a few times. Similarly, G_{-k} is mentioned many times but not clearly defined.  I would suggest to the authors to reorder the subsections in the method part to first outline the main idea: (parallel generators to capture different parts of overall distribution), mention the need to prevent redundancy between the generators (C), and mention some technical overhead in determining how to process all generated images by D. All of this may be discussed within the context of Fig 1. Also Fig 1a-b may be combined and may aid in explanation.

Experiments - Comparison is limited to single generator models. Many other generator approaches exist beyond a single generator/discriminator GAN. In particular, different loss functions for training the generator (LS-GAN etc). Missing some relevant details like why use HogWild or what it is.

Minimal understanding - I would like to know what exactly each generator contributes in the real world datasets. Can you show some generations from each mode? Is there a human perceivable difference?

Figure 4: why does the inception score for the single generator models vary with the #generators?

Last paragraph before 4.2.1: Please clarify this sentence - “we designed a relatively strong discriminator with a high learning rate, since the gradient vanish problem is not observed in reverse KL GAN.”

Typo: last line page 7: “we the use” → “we use the”

---

### Official Review · AnonReviewer1 · 2017-11-28
**Promising direction, but needs more work**

**Rating:** 5
**Confidence:** 4

**Review:**

The paper proposes to use multiple generators to fix mode collapse issue. The multiple generators are trained to be diverse. Each generator uses the reverse KL loss so that it models a single mode. One disadvantage is that it increases the number of networks (and hence the number of parameters).

The paper needs some additional experiments to convincingly demonstrate the usefulness of the proposed method. Experiments on a challenging dataset with large number of classes (e.g.  ImageNet as done by AC-GAN paper) would better illustrate the power of the method.

AC-GAN paper:
Conditional Image Synthesis with Auxiliary Classifier GANs
https://arxiv.org/pdf/1610.09585.pdf

The paper lacks clarity in some places and could use another round of editing/polishing.

---

### Public Comment · (anonymous) · 2017-11-22
**Generative Adversarial Parallelization**

[1] Generative Adversarial Parallelization (GAP) is framework where multiple generative and discriminator are trained simultaneously via exchanging their discriminators, which eliminates the tight coupling between a generator and discriminator.  Is their relationship between the method proposed and GAP?

[1] Daniel Jiwoong Im, He Ma, Chris Dongjoo Kim, Graham Taylor. Generative Adversarial Parallelization https://arxiv.org/abs/1612.04021

---

> ### Author Response · Authors · 2017-11-26
> **Re: Generative Adversarial Parallelization**
>
> Hello, thanks for your comment. In GAP, multiple discriminators are trained and the swap operator will reduce the coupling between a generator and discriminator since a tight pair could lead to mode collapse problem. However, in our proposed method, only one global discriminator is used and each generator is trained to capture different modes of the data distribution. The extra component C will penalize those generators that collapse to the same mode. Another simple understanding of our proposed method is that each generator tries to capture the data distribution while keeps a distance with any other generators. So that the search space will be partitioned into k separate parts(k is the number of generator) and each generator will capture a certain part.
> So our method is different from GAP, where GAP use swap operator to bring different adversaries to each generator, while in our method, we partition the space using extra component C, and each generator will capture a certain part of the data distribution.

---

### Decision · Program_Chairs · 2018-01-29
**ICLR 2018 Conference Acceptance Decision**

**Decision:**

Reject

**Comment:**

The paper aims to address the mode collapse issue in GANs by training multiple generators and forcing them to be diverse.

Reviewers agree that the proposed solution is not novel and has disadvantages such as increased parameters due to multiple generator models. The authors do not provide convincing arguments as to why the proposed approach should work well. The experiments presented also fail to demonstrate this. The results are limited to smaller MNIST and CIFAR10 datasets. Comparisons with approaches that directly address the mode collapse problem are missing.